# SynSQL: Synthetic Database Generation for Robust Evaluation of Text-to-SQL Systems

## Abstract

A central challenge in test-time scaling for text-to-SQL is generating test databases that can reliably validate arbitrary queries, yet existing tools remain narrow in scope and limited in capability. We introduce SynSQL, a framework for synthesizing test databases conditioned on natural language questions and schema structure. Unlike prior approaches that generate data from gold queries, SynSQL leverages large language models to generate tables directly from question–schema alignment, while remaining compatible with gold queries when available for evaluation. The framework consists of a schema selector, a synthesizer, and a critic with iterative refinement, which jointly align semantic cues from the question with structural constraints from the schema to guide database synthesis. Experiments on the Spider and BIRD benchmarks demonstrate that SynSQL produces realistic, constraint-respecting databases that effectively stress-test text-to-SQL models. SynSQL not only complements the coverage of human-curated benchmarks but also outperforms prior test database generation methods across diverse schema complexities. On Spider, SynSQL achieves a 93.04% success rate, surpassing the original human-authored dataset (92.55%), and on BIRD it attains a 79.23% agreement rate, substantially higher than prior automated methods, all while operating without access to gold queries during data generation.

## 1 Introduction

Verifying program correctness, ensuring that a system behaves as intended, has long been a central challenge in computer science, complicated by the undecidability of the halting problem (Turing, 1936). Foundational work in formal methods sought principled ways to reason about correctness (Hoare, 1969), but scaling such approaches to modern machine learning systems remains elusive. In practice, benchmarks have emerged as practical surrogates for correctness: carefully curated collections of test data that enables systematic evaluation and comparison across systems. From early datasets such as MNIST (LeCun et al., 2002) to large-scale language understanding benchmarks like SQuAD (Rajpurkar et al., 2016), GLUE (Wang et al., 2018), and HELM (Liang et al., 2022), benchmarks have repeatedly provided the common ground on which progress in AI research is measured.

In text-to-SQL, benchmarks such as Spider (Yu et al., 2018) and BIRD (Li et al., 2023) have played this role with significant impact. By pairing natural language (NL) questions with SQL queries over curated relational databases, these benchmarks have enabled rapid iteration and comparison of text-to-SQL systems. However, despite their central role, recent analyses have shown that benchmark-driven evaluation remains fragile, narrow in scope, and often misaligned with real deployment needs (Mitsopoulou & Koutrika, 2025; Renggli et al., 2025).

A core challenge lies in the quality of databases themselves. Since benchmarks rely on fixed, static databases, any inconsistencies or artifacts directly undermine evaluation reliability. Issues such as referential integrity violations, unexpected `NULL` values, case mismatches between questions and database content, or noisy entries can lead to misleading outcomes: false positives (incorrect queries that nevertheless return the expected result) or false negatives (semantically equivalent queries that yield different outputs). Consequently, the same query may be judged differently depending on the database (Zhong et al., 2020).

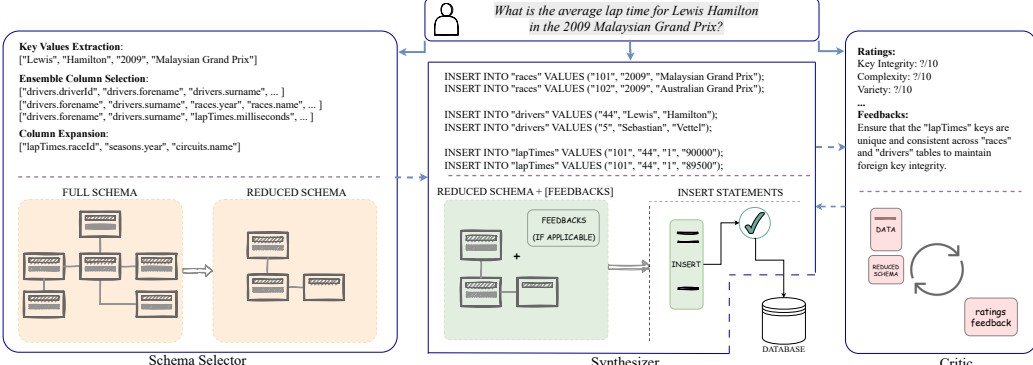

Figure 1: Overview of the SynSQL framework. The schema selector identifies relevant schema elements and reduces the schema space. The synthesizer generates test data based on the NL question and reduced schema. The critic evaluates the quality of the generated data and provides feedback for improvement.

Beyond benchmark construction, evaluation often takes place in settings where suitable databases do not exist. During system development, teams face cold-start scenarios in which schemas exist but no database content is available, or privacy-sensitive applications where real data cannot be shared (Ge et al., 2024). In such cases, developers are left with fragile proxy metrics or ad hoc synthetic data, neither of which provides robust evaluation. Addressing these challenges requires frameworks capable of synthesizing realistic, constraint-respecting test databases tailored for text-to-SQL evaluation.

**Limitations of Prior Work** Existing approaches for test database generation, such as AGENDA (Deng et al., 2005) and Qex (Veanes et al., 2010), as well as more recent systems like XData (Chandra et al., 2015; Somwase et al., 2024) and TestSuiteAccuracy (Zhong et al., 2020), focus on generating input tables to test SQL queries. These methods assume access to a gold SQL query $q$, presumed correct, and generate neighboring or mutated queries that are syntactically similar but semantically incorrect. Test data is then designed to differentiate $q$ from its mutations. While effective in query-centric testing, this paradigm is bounded by the complexity of queries the framework can handle and does not generalize to settings when evaluation must be driven directly by natural language questions and schema structure.

**Our Approach** In contrast to prior work, we explore a more realistic and challenging setting: test database generation directly from the NL question and schema. This removes the dependency of the data synthesis process on curated SQL annotations and their mutations, enabling broader applicability. Our central research question is: *Can a meaningful test database be constructed using only the NL question and schema, in order to assess the correctness of the generated SQL query?*

Our hypothesis is that semantic signals from the question, when combined with schema structure, can guide the synthesis of input tables that reveal errors, ambiguities, or misinterpretations in SQL generation. To this end, we introduce SynSQL, a framework for automatic databases synthesis that leverages large language models (LLMs) for question understanding and systematically aligns question semantics with schema constraints. SynSQL adopts a modular, three-agent design (Figure 1): the *Schema Selector* prunes irrelevant relations to simplify schemas; the *Synthesizer* generates test tables conditioned on the NL question and reduced schema; and the *Critic* evaluates the table data and provides feedback for iterative refinement. This interactive pipeline produces realistic, minimal, and constraint-respecting databases that adapt to both schema and query semantics.

Our contributions can be summarized as follows: (1) We introduce SynSQL, a framework for synthetic database generation in text-to-SQL tasks to extend the existing evaluation methods. (2) Our framework ensures generation of targeted, minimal, and realistic databases that respect schema constraints (e.g. foreign key relationships and uniqueness) and align with question semantics to differentiate between correct and incorrect SQL queries. (3) Through comprehensive experiments

on the Spider and BIRD benchmarks, we demonstrate that SynSQL complements the coverage of human-curated benchmarks and also outperforms previous automated test database generation methods across varying schema complexities. Specifically, on Spider, SynSQL achieves a 93.04% success rate, surpassing the original human-authored databases (92.55%), and on BIRD, it reaches a 79.23% agreement rate, outperforming prior automated approaches.

## 2 RELATED WORK

**Test Data Generation for SQL**   Test data generation for SQL queries has been explored through various strategies, including random data generation (Bati et al., 2007), constraint-based approaches (Veanes et al., 2010; Shah et al., 2011), and mutation-based techniques (Tuya et al., 2007). Qex (Veanes et al., 2010) synthesizes tuples using SMT solving to guarantee non-empty query results.XData (Shah et al., 2011) builds on this idea by generating datasets that differentiate a given SQL query from its syntactically similar but semantically incorrect mutations, with recent follow-up work (Somwase et al., 2024) supporting more complex SQL constructs. TestSuiteAccuracy (Zhong et al., 2020), introduced in the context of the Spider benchmark (Yu et al., 2018), employs fuzzing techniques(Padhye et al., 2019) to create large numbers of random databases, from which a small subset is selected to maximize code coverage with respect to the gold query. In contrast, APEL (Zhong et al., 2022) synthesizes small human-interpretable input datasets to enable non-programmers to reliably label program outputs.

**Synthetic Data Generation with LLMs**   LLMs have been increasingly used for for generating synthetic data across NLP tasks (Wang et al., 2022; Ramesh et al., 2022; Nadăs et al., 2025; Yoo et al., 2021; Dai et al., 2025), including program test synthesis (Wang et al., 2024; Chen et al., 2022; Yuan et al., 2023; Bhatia et al., 2024; Yang et al., 2024). In the domain of text-to-SQL, recent work has focused on producing large synthetic NL–SQL corpora for supervised learning, such as Gretel (Meyer et al., 2024) and Omni-SQL (Li et al., 2025). Complementary work also explores question generation over structured sources using LLM reasoning strategies (Liang et al., 2023), though these methods target question synthesis rather than database construction. Beyond data generation, LLMs have also been used in database-related tasks such as SQL dialect translation and query adaptation (Ngom & Kraska, 2024; Daviran et al., 2025). These efforts demonstrate the versatility of LLMs in understanding schema structure and SQL semantics but do not address the challenge of creating executable, constraint-respecting test databases.

**Relation to Our Work**   Unlike prior work that focuses on generating NL–SQL pairs or mutating SQL queries, our framework targets the synthesis of complete test databases directly from question–schema alignment. SynSQL's three-stage design—schema selection, LLM-guided data synthesis, and iterative criticism—enforces schema integrity and semantic alignment, enabling realistic and compact databases tailored for evaluating text-to-SQL systems.

## 3 METHODOLOGY

### 3.1 PROBLEM FORMULATION

Given a natural language question $Q$, a database schema $S$, and optional auxiliary knowledge $K$, the task of synthetic data generation using a language model $\pi$ can be formulated as:

$$D_{syn} = f(Q, S, K; \pi) \tag{1}$$

where $D_{syn}$ denotes the generated synthetic database and $f$ is a generation function parameterized or guided by the language model $\pi$. When $D_{syn}$ is used to evaluate the correctness of a predicted SQL query $q_{pred}$ against a reference query $q_{gold}$, it must be populated with realistic rows that reflect the semantic cues of $Q$; for example, if $Q$ asks for *What was Brent Thomason's major?*, the database should contain name of majors for student named Brent Thomason. In addition, $D_{syn}$ should enable clear differentiation between correct and incorrect queries by producing different results for

semantically distinct queries, while also respecting all schema constraints such as foreign keys and domain-specific rules.

## 3.2 Overview of Framework

We introduce SynSQL, a three-staged framework for generating high-quality synthetic databases tailored for evaluating text-to-SQL systems. The pipeline consists of a Schema Selector, a Synthesizer, and a Critic. As shown in Figure 1, SynSQL incorporates a feedback loop between the synthesizer and the critic. After an initial round of data generation, the critic evaluates the output and provides structured feedback. If the data achieves a score above a predefined threshold (e.g., 8 out of 10), the process terminates. Otherwise, the synthesizer incorporates the feedback into a new round of data generation. This loop typically continues for a fixed number of rounds or until the generated data satisfies all quality criteria.

## 3.3 Schema Selection

Real-world database schemas often contain many tables and columns (sometimes in the range of hundreds), while most NL questions reference only a small subset. Providing the full schema to the model can lead to the generation of unnecessary or inconsistent data (e.g., mismatched column counts or malformed tables). The challenge is to accurately identify the relevant schema elements while preserving enough context for realistic and coherent data synthesis.

Our schema selection strategy differs from that of state-of-the-art text-to-SQL systems (Talaei et al., 2024; Pourreza et al., 2024; Gao et al., 2024), which typically map question keywords to schema elements (e.g., table and column names) and database values in order to prune the schema down to only those components strictly necessary for query generation. In our setting, however, no real database content is available, and the data itself must be generated synthetically. Consequently, SynSQL prioritizes recall over minimality: rather than restricting to the smallest matching subset, we retain all schema elements that could plausibly support realistic data synthesis, including those not explicitly mentioned in the question, such as foreign keys or related attributes.

---

**Algorithm 1** Schema Selector for SynSQL

---

**Require:** Natural language question $Q$, full schema $S$, auxiliary knowledge $K$, model $\pi$
 1: **Ensure:** Reduced schema $S_{reduced}$ relevant to $Q$
 2: Initialize $S_{core} \leftarrow \emptyset$
 3: **for** temperature $t$ in $\{0, 0.3, 0.7\}$ **do**
 4:     Query $\pi$ with $(Q, S, K)$ at temperature $t$ to extract core elements
 5:     $S_{core} \leftarrow S_{core} \cup$ elements returned by $\pi$
 6: **end for**
 7: Query $\pi$ for semantically related columns to $S_{core}$
 8: $S_{aux} \leftarrow$ related columns returned by $\pi$
 9: $S_{reduced} \leftarrow S_{core} \cup S_{aux}$
10: **return** $S_{reduced}$

---

To identify this relevant subset, we employ an ensemble expansion strategy. As outlined in Algorithm 1, the LLM is queried multiple times at different temperature settings to capture a diverse set of core columns. The union of the results is taken to maximize recall. Next, we expand the set by querying the LLM for semantically similar or functionally related columns. This helps in discovering auxiliary schema elements that support better data realism. At the end of this process, we have a reduced schema that includes only the relevant elements needed to cover the question. This reduced schema is then used by the data synthesizer to generate test data.

The benefit of this approach is that we can generate smaller, more focused databases that are easier to inspect and validate, while still capturing the necessary complexity to effectively evaluate SQL queries. These compact synthetic databases especially enable human-in-the-loop evaluation in cold-start development or privacy-sensitive environments, where real data is unavailable or restricted. Though not the focus of this work, SQL query correctness can then be determined by comparing

the results against expected outputs, observed by the human evaluator, even in the absence of gold queries or large-scale benchmarks.

## 3.4 DATA SYNTHESIS

Existing approaches in database synthesis focus on techniques that generate data to differentiate between reference SQL query and its neighbors/mutations. This process is bounded by the complexity of queries their framework can handle. If the system cannot yet process a certain query structure, then it also cannot generate databases that can kill mutations for that structure. For example, in XData (Somwase et al., 2024), a feature like `ORDER BY` falls outside its scope. Or in TestSuiteAccuracy (Zhong et al., 2020), the authors note that too many `WHERE` operations can lead to ineffective distinguishing of neighboring queries.

In contrast, our synthesizer does not rely on reference queries and focuses on generating realistic databases that align with the semantics of the NL question. We leverage LLMs to generate SQL INSERT statements that can populate the reduced schema with synthetic yet realistic data. The generation process is guided by the intent of the NL question and aims to produce data that both aligns with the semantics and provides meaningful contrast between correct and incorrect SQL queries.

**Data Validation** Database schemas typically encode structural constraints, including primary and foreign keys, uniqueness, and value ranges. Generating synthetic data that satisfies these constraints, while preserving realistic value distributions, is challenging. Failure to respect these constraints can result in data that causes SQL queries to fail or produce misleading results. Once the initial data is generated, we apply rule-based postprocessing to ensure correctness and schema compliance. This involves removing tables or columns present in data but not in the schema, only retaining data that respects schema constraints, enforcing case sensitivity based on named entities and literals extracted from the question, and ensuring that the number of values matches the number of columns in each table. When discrepancies are found, we either pad with `NULL`s or truncate values as needed. This step is critical for maintaining the syntactic and semantic integrity of the SQL statements.

## 3.5 DATA CRITIC

The synthetic data produced by the synthesizer may not always meet the desired quality standards. There could be issues such as misalignment with the NL question, violations of schema constraints, or lack of edge cases in the data distribution. Inspired by the idea of self-correction for LLMs (Pan et al., 2023), we introduce a Critic module that evaluates the generated data and provides feedback for improvement which determines whether the data should be accepted or refined. It scores the data on a scale of 1 to 10 across several dimensions, including alignment with question hints, key and referential integrity, schema coverage, complexity of the data, variety in records, and overall relevance to the question. Instead of using simple pass/fail rules, the critic provides a detailed evaluation that allows the synthesizer to improve in targeted ways. If the average score meets the quality threshold, the data is accepted. Otherwise, the critic's feedback is incorporated into a new iteration of data generation. This loop ensures that the final dataset not only conforms to the schema and executes without errors but also meaningfully tests the correctness of SQL predictions in the context of the original question.

## 4 EXPERIMENTS

Our experimental evaluation is designed to assess two core aspects: (1) the quality of the synthetic data generated by SynSQL compared to human-curated datasets and competitive automated approaches, and (2) the contribution of different components of SynSQL via an ablation study.

## 4.1 EXPERIMENTAL SETUP

**Datasets** We evaluate SynSQL on two widely used text-to-SQL benchmarks: Spider (Yu et al., 2018) and BIRD (Li et al., 2023). Spider features natural language questions paired with SQL queries over relatively simple schemas, while BIRD includes more complex queries involving multiple joins, nested subqueries, and advanced aggregations. This contrast enables a comprehensive

assessment of SynSQL across varying schema complexity and linguistic difficulty. Our auxiliary knowledge K (as defined in Section 3.1) is set to the evidence or hints associated with each question for BIRD, and to an empty string for Spider, which does not provide such hints.

**Baselines** We compare SynSQL to three baselines: (1) **BIRD Original** and (2) **Spider Original**, which refer to the original human-authored databases included in the dev splits of the respective benchmarks and serve as the gold standard for test data; and (3) **TestSuiteAccuracy (TSA)** by Zhong et al. (2020) that generates test databases by finding neighboring queries to each gold query and applying fuzzing-based techniques to produce random databases, benefiting from access to the gold query during data generation. Among existing automated test data generation methods, TSA is the only approach that has an available codebase and can be applied to any standalone dataset such as Spider or BIRD, making it the only directly comparable baseline for our setting.

**Text-to-SQL Systems** To evaluate the ability of SynSQL generated databases to separate correct queries from incorrect ones, we use two competitive text-to-SQL models to generate candidate SQL queries: **DIN-SQL** (Pourreza & Rafiei, 2023) and **DAIL-SQL** (Gao et al., 2023). The predictions produced by these models are evaluated against the gold queries using both human-curated and synthetic databases.

### 4.2 EVALUATION METRICS

We employ three complementary metrics to assess the quality and utility of the generated databases:

**Success Rate (SR)** Using this metric we measure the fraction of questions for which the gold SQL query produces a non-empty result on the test database. This indicates whether the generated data captures the semantic intent of the natural language question (from the perspective of the human who wrote the gold query). Random data often fails here, so aligning with question intent is crucial.

**Execution Accuracy (EX)** We can evaluate the fraction of questions for which the model-generated SQL query produces the same result as the gold SQL query when executed on the test database (Zhong et al., 2017). The comparison of this metric with human-curated databases, measures the database's ability to distinguish between correct and incorrect SQL queries.

**Agreement Rate (AR)** Inspired by Cohen's Kappa (Cohen, 1960) score, this metric assesses the level of agreement between the discriminative power of model-generated and human-curated databases on a query-by-query basis. Using this metric we can show that if the evaluation of each query on the synthetic database agrees with the evaluation on the original database, then the synthetic database is in agreement with the original one. Formally, AR is defined as:

$$\text{AR} = \frac{P_o - P_e}{1 - P_e} \tag{2}$$

where $P_o$ is the observed agreement (i.e., the proportion of queries where both databases yield the same evaluation result) and $P_e$ is the expected agreement by chance, calculated based on the marginal probabilities of each database's evaluations. For more on this metric please refer to Appendix section A.2.

## 5 RESULTS

### 5.1 MAIN RESULTS

We present the performance of the proposed SynSQL framework using a mix of proprietary and open-source language models: GPT-4.1-mini, Gemini-2.5-Flash, and Qwen-3-8B. The critic module uses a maximum of three refinement iterations, terminating early if the data achieves a quality score of 8.0 or higher on a 10-point scale. For our vanilla baseline, we prompt the LLM to generate synthetic data in a single pass without schema reduction or critic feedback. The vanilla baseline is equivalent to SynSQL with only the synthesizer component, using the same prompting strategy but operating on the full schema without iterative refinement. For more details on implementation, please refer to section 6.

Table 1: Performance comparison of SynSQL and baseline methods on BIRD and Spider dev sets. SR: success rate (%), AR: agreement rate (%), EX: execution accuracy (%), and performance gap ($\Delta$) is defined as $|EX_{orig} - EX_{method}|$.

| Dataset | Method | SR | DAIL-SQL | | | DIN-SQL | | |
|---|---|---|---|---|---|---|---|---|
| | | | AR↑ | EX | $\Delta\downarrow$ | AR↑ | EX | $\Delta\downarrow$ |
| BIRD | BIRD (Original) | 99.87 | 100.00 | 52.93 | - | 100.00 | 41.39 | - |
| | Vanilla GPT-4.1-Mini | 69.43 | 58.57 | 68.12 | 15.19 | 61.88 | 55.61 | 14.22 |
| | Vanilla Gemini-2.5-Flash | 67.14 | 61.56 | 66.95 | 14.02 | 65.64 | 54.95 | 13.56 |
| | SynSQL(Qwen-3-8B) | 73.60 | 70.64 | 58.21 | 5.28 | 72.94 | 47.78 | 6.39 |
| | SynSQL(Gemini-2.5-Flash) | 80.57 | **75.46** | 55.61 | 2.68 | **79.23** | 44.00 | 2.61 |
| | SynSQL(GPT-4.1-Mini) | **82.07** | 73.51 | 55.02 | **2.09** | 78.90 | 43.09 | **1.70** |
| Spider | Spider (Original) | 92.55 | 100.00 | 80.66 | - | 100.00 | 80.46 | - |
| | Vanilla GPT-4.1-Mini | 91.88 | 78.20 | 83.07 | 2.41 | 76.44 | 82.85 | 2.39 |
| | Vanilla Gemini-2.5-Flash | 82.59 | 74.33 | 84.24 | 3.58 | 72.46 | 83.08 | 2.62 |
| | SynSQL(Qwen-3-8B) | 77.18 | 66.99 | 84.24 | 3.58 | 66.89 | 81.72 | 1.26 |
| | SynSQL(Gemini-2.5-Flash) | 92.84 | **80.69** | 81.72 | 1.06 | **76.53** | 81.33 | 0.87 |
| | SynSQL(GPT-4.1-Mini) | **93.04** | 78.05 | 81.24 | **0.58** | 76.01 | 79.79 | **0.67** |

Table 2: Performance comparison of SynSQL and TestSuiteAccuracy (TSA) alongside the original benchmarks. BIRD results are computed on a subset of 922 examples where TSA evaluation was feasible.

| Dataset | Method | SR | DAIL-SQL | | | DIN-SQL | | |
|---|---|---|---|---|---|---|---|---|
| | | | AR↑ | EX | $\Delta\downarrow$ | AR↑ | EX | $\Delta\downarrow$ |
| BIRD* | BIRD (Original) | 99.57 | 100.00 | 51.74 | - | 100.00 | 38.72 | - |
| | TSA Zhong et al. (2020) | 77.76 | 60.97 | 57.27 | 5.53 | 64.71 | 46.31 | 7.59 |
| | SynSQL(Gemini-2.5-Flash) | 81.67 | **74.76** | 54.56 | 2.82 | 76.14 | 41.97 | 3.25 |
| | SynSQL(GPT-4.1-Mini) | **82.86** | 70.85 | 53.90 | **2.16** | **76.45** | 40.67 | **1.95** |
| Spider | Spider (Original) | 92.55 | 100.00 | 80.66 | - | 100.00 | 80.46 | - |
| | TSA Zhong et al. (2020) | **99.03** | **83.15** | 76.31 | 4.35 | **79.99** | 75.73 | 4.73 |
| | SynSQL(Gemini-2.5-Flash) | 92.84 | 80.69 | 81.72 | 1.06 | 76.53 | 81.33 | 0.87 |
| | SynSQL(GPT-4.1-Mini) | 93.04 | 78.05 | 81.24 | **0.58** | 76.01 | 79.79 | **0.67** |

**Success Rate.** As shown in Table 1, SynSQL with GPT-4.1-Mini as the base model, achieves a success rate of 82.07% on BIRD, outperforming both the vanilla GPT-4.1-Mini baseline (69.43%) and other LLM configurations. On Spider, SynSQL again leads with 93.04%, even surpassing the original human-authored database (92.55%). This is partly due to inconsistencies in the Spider original databases, such as missing data or formatting issues (e.g., trailing spaces), which SynSQL avoids by design. Appendix figures 18 and 19 provide examples of such issues.

**Agreement Rate.** SynSQL achieves high agreement rates with the original databases, indicating that it effectively preserves and complements the evaluation characteristics of human-curated data. On BIRD, SynSQL(Gemini-2.5-Flash) attains AR scores of 75.46% for DAIL-SQL and 79.23% for DIN-SQL, significantly outperforming the vanilla GPT-4.1-Mini baseline (61.56% and 65.64%, respectively). On Spider, SynSQL again achieves substantial agreement scores, closely matching the original database's performance.

**Execution Accuracy and Gap.** On BIRD, the EX gap between SynSQL (GPT-4.1-Mini) and the original data is just 2.09 for DAIL-SQL and 1.70 for DIN-SQL, an order of magnitude smaller than the gaps observed for vanilla GPT-4.1-Mini (15.19 and 14.22, respectively). On Spider, all methods perform more closely due to its simpler schema and queries. SynSQL again performs competitively with EX gaps under 1.0. Consequently, SynSQL closely preserves the ranking of text-to-SQL models observed with the original databases. For instance, DAIL-SQL consistently outperforms DIN-SQL across all SynSQL configurations, closely mirroring the execution gaps seen with the original data. This consistency in model ranking further validates the effectiveness of SynSQL-generated databases for complementary robust evaluation.

**Comparison with TestSuiteAccuracy (TSA).** As shown in Table 2, while TSA performs well on Spider, achieving the highest success rate at 99.03%, its effectiveness drops on BIRD, with a success rate of 77.76%, significantly lower AR of 60.97% for DAIL-SQL, and larger EX gaps (e.g., 5.53 for DAIL-SQL). In contrast, SynSQL achieves higher success (82.86%), higher agreement rate (74.76%), and lower EX gaps (2.16), indicating better semantic alignment and discriminative power on complex schemas.

TSA's reliance on gold SQL queries offers an advantage on simpler benchmarks but becomes a liability on datasets like BIRD. For example, it fails to resolve foreign key references in databases like `european_football_2`, and generates impractically large test databases (e.g., multiple GBs per question) in `card_games` and `codebase_community`. Consequently, we excluded such problematic cases and report results on a filtered subset of 922 BIRD questions. For fairness, the same subset was used to evaluate SynSQL and the BIRD original databases in Table 2. Overall, SynSQL not only outperforms TSA on BIRD but also delivers stable performance across datasets, generating minimal and realistic test databases. For more on realism and minimalistic nature of these databases, see A.9. These results demonstrate SynSQL's practicality and robustness for realistic text-to-SQL evaluation scenarios.

**Performance Varying Schema Complexity** We also analyze SynSQL's performance rates across question-schema complexity levels on the BIRD dev set, defined by the total number of columns across all tables referenced in each gold query: Low (1–15), Medium (16–60), and High (61+). As shown in Appendix Figure 3, SynSQL consistently outperforms vanilla GPT-4.1-mini, with the largest margin at high complexity, 76.99% vs. 36.15% for success rate and 71.07% vs. 33.19% for agreement rate. This demonstrates SynSQL's robustness as schema complexity increases.

## 5.2 FINE-GRAINED ANALYSIS OF EXECUTION ACCURACY

To better understand the impact of synthetically generated data on execution accuracy, we conducted an error analysis comparing the outputs of DAIL-SQL queries executed on the original BIRD dev set versus the SynSQL-generated data. We randomly sampled 250 questions from the BIRD dev set and executed the corresponding DAIL-SQL queries on both the original BIRD and the SynSQL-generated databases. Comparing the outputs to the ground truth, we found 35 instances of disagreements where the evaluation results differed across the two datasets. Of these, SynSQL produced correct (positive) results in 21 cases where BIRD did not, resulting in a higher execution accuracy. Conversely, 14 cases were evaluated as positive on the original BIRD dataset but negative on SynSQL.

A closer examination revealed that 15 out of the 35 disagreements are attributed to data quality issues in the original BIRD dev set. For instance, in the `toxicology` database, the `molecule_id` column in the `bond` table is a foreign key referencing the `molecule` table. However, 101 rows in the `bond` table contain `molecule_id` values not present in the `molecule` table, causing otherwise correct queries to fail (e.g., question 286). Similarly, in the `thrombosis_prediction` database, the `ID` column in the `Examination` table is a foreign key referencing the `Patient` table, yet 694 rows in `Examination` have `ID` values absent from `Patient`, leading to possible false negatives (e.g., question 1273). These are examples of one of the key advantages of SynSQL: by generating data

Table 3: Analysis of evaluation disagreements between SynSQL and BIRD from a sample of 250 questions with 35 discrepancies.

| BIRD → SynSQL | Change |
|---|---|
| Negative → Positive | 21 |
| From False Negative to True Positive | 10 |
| From True Negative to False Positive | 11 |
| Positive → Negative | 14 |
| From False Positive to True Negative | 5 |
| From True Positive to False Negative | 9 |

that respects all schema constraints, it can avoid false negatives caused by such data quality issues in human-curated datasets.

Another example is question 357: *What type of promotion is of card 'Duress'?* The gold query includes a `NOT NULL` condition for `promoTypes`, while the predicted query does not. On the original BIRD database, this leads to false negative due to the presence of many `NULL` artifacts. However, since the question and evidence do not specify the presence of `NULL` values, the predicted query is arguably correct. SynSQL-generated database has a question-oriented design which gener-

ates data that aligns with the semantic signals of the question, resulting in a more accurate evaluation of such cases.

Table 3 summarizes the transitions in execution results on a sample of 250 questions. Notably, SynSQL corrects 10 false negatives and 5 false positives. These findings suggest that SynSQL's constraint enforcement and question-driven synthesis approach helps mitigate common data quality issues found in human-curated datasets, such as referential integrity violations and unexpected NULL values, resulting in more accurate and reliable evaluation of text-to-SQL systems on cases affected by such inconsistencies.

## 5.3 ABLATION STUDIES

Table 4 shows the effect of schema selection on performance. The SynSQL method with ensemble-expansion outperforms all ablated versions, confirming that both phases contribute meaningfully to success rate, especially on BIRD, where complex schemas increase the difficulty of accurate column selection. Using the oracle schema yields highest success rate, indicating further improvements in schema selection could enhance performance. We also observe that the average column count selected by the schema selector is significantly lower than the full schema, demonstrating SynSQL's ability to generate compact databases while maintaining high success rates. However, aggressive reduction risks omitting columns required by gold queries, causing otherwise correct queries to fail. This highlights the inherent tension between minimizing schema complexity and preserving query executability. Despite this limitation, SynSQL's ensemble-expansion strategy achieves a balance that maintains high success rates while generating significantly more compact databases than the full schema.

As illustrated in Appendix figures 16 and 17, these compact synthetic databases are easier to inspect and validate, facilitating future directions such as human-in-the-loop evaluation and generating expected outputs via table reasoning.

Table 4: Ablation study on schema selection strategies in SynSQL. We report success rate (SR) and average column count (CC) selected by the schema selector on the BIRD and Spider dev sets. All variants use GPT-4.1-Mini as both the base model and the critic.

| Method | BIRD | | Spider | |
|---|---|---|---|---|
| | SR (%) | CC | SR (%) | CC |
| SynSQL w Oracle Schema | 91.46 | 4.71 | 94.58 | 2.85 |
| SynSQL | 82.07 | 8.37 | 93.04 | 6.71 |
| SynSQL w/o Expansion | 79.53 | 5.42 | 92.75 | 3.92 |
| SynSQL w/o Ensemble-Expansion | 77.38 | 4.99 | 91.88 | 3.56 |
| SynSQL w/o Schema Selection (Full-Schema) | 71.25 | 75.56 | 92.94 | 24.55 |

As illustrated in Figure 2, the impact of the critic component on the success rate of SynSQL across three LLMs on the BIRD dev set is significant. Incorporating the critic consistently improves performance: Qwen-3-8B improves from 67.86% to 73.60%. The effect is less pronounced for GPT-4.1-Mini, which already performs strongly, but the critic still ensures more stable and reliable results. These results demonstrate that the critic plays a key role in improving the quality of synthetic data. By enforcing alignment with the question intent, schema constraints, and data diversity, the critic enables the synthesizer to produce more accurate and executable SQL queries. This

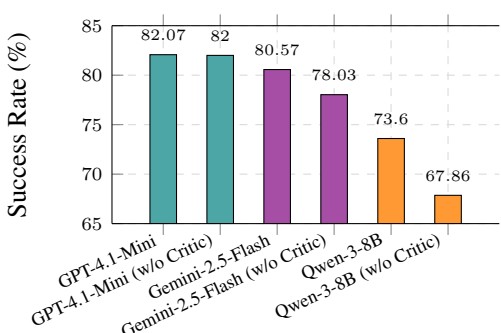

Figure 2: Impact of the critic component on success rate (%) of SynSQL with three different LLMs on the BIRD dev set.

iterative feedback loop is particularly valu-
able for smaller or weaker models, which are more prone to generating invalid or incomplete outputs.

# 6 CONCLUSION

In this work, we presented SynSQL, a novel framework for synthetic database generation in text-to-SQL evaluation, producing databases that respect schema constraints and align with the natural language question intent. Extensive experiments on Spider and BIRD benchmarks show that SynSQL not only complements human-curated datasets but also outperforms existing automated methods, particularly as schema complexity increases. These results highlight SynSQL's ability to enhance the effectiveness and reliability of text-to-SQL evaluation. The realistic and minimal nature of the synthetic data generated by SynSQL, combined with its ability to address key challenges in current evaluation practices, such as referential integrity violations or unexpected and noisy artifacts, paves the way for more robust and scalable evaluation of text-to-SQL systems encountered at real-world Text-to-SQL challenges.

## REPRODUCIBILITY STATEMENT

The link to the anonymous codebase for SynSQL is available in the README.md file in supplementary materials. SynSQL is implemented in Python using the LangChain framework (Chase, 2022). For data generation, we employ a mix of proprietary and open-source language models: GPT-4.1-mini (OpenAI, 2023), Gemini-2.5-Flash, and Qwen-3-8B. Each experiment uses a single model as the base for the schema selector, synthesizer, and critic. Qwen-3-8B experiments were run on a server with NVIDIA A100 GPUs (40GB RAM), while GPT-4.1-mini and Gemini-2.5-Flash were accessed via their respective APIs. For the TestSuiteAccuracy (TSA) baseline, we use the official implementation provided by Zhong et al. (2020) and a server equipped with an AMD EPYC 7601 32-Core Processor and 1TB RAM.

## ETHICS STATEMENT

Both datasets used in our experiments, Spider and BIRD, are publicly available and widely used benchmarks in the text-to-SQL research community. We have ensured that our use of these datasets complies with their respective licenses and terms of use. The synthetic data generated by SynSQL is created solely for research purposes and does not contain any personally identifiable information or sensitive content. The language models employed in our framework, including GPT-4.1-mini, Gemini-2.5-Flash, and Qwen-3-8B, are accessed through their respective APIs or open-source implementations. We adhere to the usage policies and guidelines set forth by the providers of these models to ensure ethical use. We also acknowledge the potential bias inherent in large language models, which may inadvertently influence the synthetic data generation process. This falls within the broader challenges of bias in AI and LLMs. Additionally, in accordance with the ICLR 2026 Code of Ethics, we acknowledge the use of large language models to assist with the polishing of the writing in this paper.

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

## A  APPENDIX

### A.1  SUCCESS AND AGREEMENT RATES BY SCHEMA COMPLEXITY

We define schema complexity levels based on the number of columns involved in the gold SQL query. Specifically, we count the total number of columns across all tables referenced in each gold query; higher column counts generally correlate with more complex joins, filters, and reasoning steps. Based on the distribution of complexity levels in the BIRD dataset, we define three buckets: **Low Complexity:** Questions with a total column count of 1-15. **Medium Complexity:** Questions with a total column count of 16-60. **High Complexity:** Questions with a total column count of 61 or more.

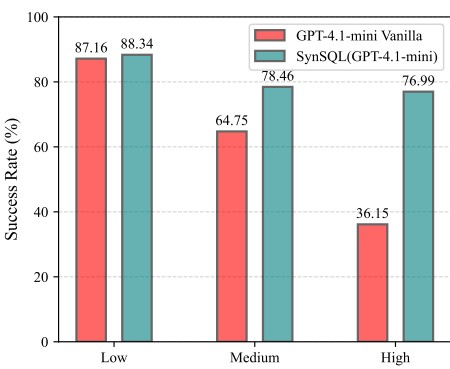 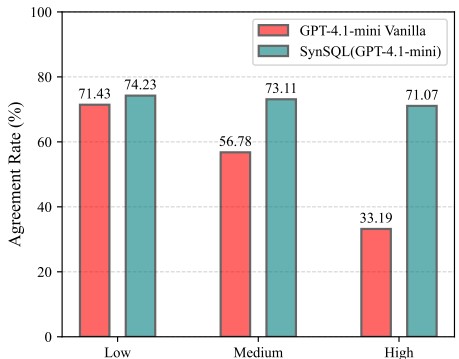

Figure 3: Success and agreement rates of SynSQL vs. GPT-4.1-Mini on BIRD dev set, broken down by schema complexity (Low: 1-15 columns, Medium: 16-60 columns, High: 61+ columns).

## A.2 AGREEMENT RATE METRIC DETAILS

The Agreement Rate (AR) metric is inspired by Cohen's Kappa score Cohen (1960), which measures inter-rater reliability. In our context, we treat the original human-curated database and each synthetic database as two "raters" that evaluate the correctness of SQL queries generated by text-to-SQL models. To compute AR, we first execute each model-generated SQL query on both the original and synthetic databases. Each execution yields a binary outcome: correct (the query produces the expected result) or incorrect (it does not). We then construct a confusion matrix based on these outcomes, counting the number of queries that fall into each of the four possible categories: True Positive (TP): Both databases evaluate the query as correct. True Negative (TN): Both databases evaluate the query as incorrect. False Positive (FP): The original database evaluates the query as incorrect, but the synthetic database evaluates it as correct. False Negative (FN): The original database evaluates the query as correct, but the synthetic database evaluates it as incorrect. Using these counts, we calculate the observed agreement $P_o$ as:

$$P_o = \frac{TP + TN}{TP + TN + FP + FN} \tag{3}$$

Next, we compute the expected agreement $P_e$ by considering the marginal probabilities of each database's evaluations:

$$P_e = \left( \frac{(TP + FP)(TP + FN)}{(TP + TN + FP + FN)^2} \right) + \left( \frac{(TN + FN)(TN + FP)}{(TP + TN + FP + FN)^2} \right) \tag{4}$$

Finally, the Agreement Rate (AR) is calculated as:

$$AR = \frac{P_o - P_e}{1 - P_e} \tag{5}$$

An AR score of 1 indicates perfect agreement between the two databases, while a score of 0 indicates no better agreement than random chance. Negative values suggest systematic disagreement. By using AR, we can quantify how well the synthetic database preserves the evaluation characteristics of the original human-curated data on a query-by-query basis. There are ranges for interpreting AR scores:

- 0.81 - 1.00: Almost perfect agreement
- 0.61 - 0.80: Substantial agreement
- 0.41 - 0.60: Moderate agreement
- 0.21 - 0.40: Fair agreement
- 0.00 - 0.20: Slight agreement
- $< 0.00$: Poor agreement

Combining AR with success rate and execution accuracy offers a more complete assessment of synthetic database quality for text-to-SQL evaluation. While execution accuracy reflects overall model performance and preserves model ranking, it does not measure per-query consistency between databases. AR addresses this by quantifying agreement on individual queries, providing finer-grained insight and mitigating potential inflation of aggregate metrics.

## A.3 ERROR ANALYSIS OF SUCCESS RATE

To better understand the performance of SynSQL in aligning with question intent, we performed error analysis on a random sample of 500 questions from the BIRD dataset. Of these, 84 questions returned an empty set as the result of the gold query, corresponding to an 83.2% success rate.

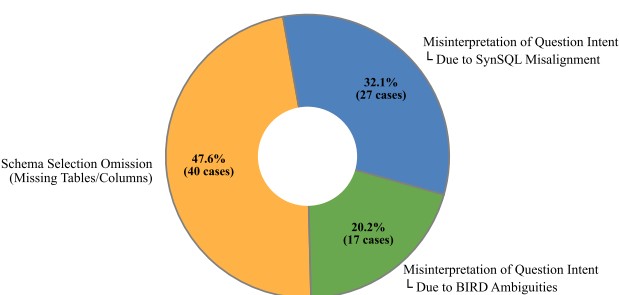

Figure 4: Breakdown of error cases in SynSQL success rate analysis (84 failures out of 500 BIRD dev questions).

Figure 4 summarizes the breakdown of failure cases and their causes. Of the 84 failed questions, 40 were due to schema selector failures. In these cases, schema reduction led to the omission of one or more tables or columns used in the gold query. This does not necessarily mean the generated data is meaningless; rather, the human annotator who wrote the gold query may have targeted different schema elements than the LLM. For example, in question 387 from the `card_games` database (Figure 5):

> **Question**: What are the cards for set OGW? State the colour for these cards.
>
> **Evidence**: set OGW refers to setCode = 'OGW'
>
> **Gold Query**: SELECT id, colors FROM cards WHERE id IN (SELECT id FROM set_translations WHERE setCode = 'OGW')

Figure 5: An example of schema selection failure. The synthetic data omits the `setCode` column from `set_translations`, leading to a failed query.

SynSQL has generated data for the `setCode` column in `cards`, but omitted the `setCode` column from `set_translations` during schema selection. The gold query joins both tables on `setCode`, leading to failure. However, the synthetic data still contains valid `setCode` values, just not in the joined table. This highlights the challenge of schema selection in open-ended text-to-SQL tasks, where multiple valid interpretations exist.

The remaining 44 failures were due to misinterpretation of question intent. For example, in question 156 from the financial database (Figure 6):

> **Question**: Who is the owner of the account with the largest loan amount?
>
> **Evidence**: N/A
>
> **Gold Query**: SELECT T1.client_id FROM disp AS T1 INNER JOIN account AS T3 ON T1.account_id = T3.account_id INNER JOIN loan AS T2 ON T3.account_id = T2.account_id WHERE T1.type = 'OWNER' ORDER BY T2.amount DESC LIMIT 1

Figure 6: Example of misinterpretation: the synthetic data contains values such as `owner` (lowercase) in the `type` column, while the gold query expects `OWNER` (uppercase). This case sensitivity mismatch leads to a failed query.

Here, the synthetic database reflects the casing found in the question or evidence, but the gold query expects a different case. Such mismatches between generated data and gold query expectations, especially regarding case sensitivity or value formatting, can result in lower success rates.

Another example is question 90 from the financial database (Figure 7):

> **Question**: How many accounts who have region in Prague are eligible for loans?
>
> **Evidence**: A3 contains the data of region
>
> **Gold Query**: SELECT COUNT(T1.account_id) FROM account AS T1 INNER JOIN loan AS T2 ON T1.account_id = T2.account_id INNER JOIN district AS T3 ON T1.district_id = T3.district_id WHERE T3.A3 = 'Prague'

Figure 7: An example of misinterpretation due to synthetic data not matching gold query conditions. The synthetic data contains values that do not satisfy the gold query's WHERE clause, leading to failure.

The gold query expects `district.A3 = 'Prague'`, but the synthetic data contains values such as `Prague 1`, `Prague 2`, and `Prague 3`. Here, the LLM generated region names with appended numbers, resulting in a mismatch with the gold query's expected value.

Some misinterpretations are due to misalignment between the question and the gold query in the BIRD dev set, rather than errors by SynSQL. For example, in question 803 from the `Superhero` database (Figure 8):

> **Question**: What is the power ID of cryokinesis?
>
> **Evidence**: power ID refers to superpower.id; cryokinesis refers to power_name = 'cryokinesis'
>
> **Gold Query**: SELECT id FROM superpower WHERE power_name = 'Cryokinesis'

Figure 8: An example of misinterpretation due to inconsistencies between question/evidence and gold query in the BIRD dev set. The synthetic data aligns with the question, but not the gold query, leading to failure.

In this case, the question and evidence refer to `cryokinesis` (lowercase), while the gold query expects `'Cryokinesis'` (capitalized). The synthetic database contains `power_name = 'cryokinesis'`, resulting in a mismatch with the gold query and subsequently lower success rate. Similarly, in question 758, the question and evidence specify `race = 'human'`, but the gold query expects `race = 'Human'`. In question 415, the question and evidence use `Status`

= 'legal', while the gold query expects `Status = 'Legal'`. The synthetic data generated by SynSQL reflects the casing found in the question, leading to mismatches with the gold query.

In summary, among the 44 misinterpretation cases, 17 stem from insufficient or ambiguous information in the BIRD dev set, while 27 are attributable to SynSQL's generation errors. The following BIRD dev set questions could not be correctly handled by SynSQL due to a lack of necessary information in the dataset for generating appropriate synthetic data. Such cases are likely to be challenging for any text-to-SQL system: 22, 73, 180, 309, 415, 758, 769, 803, 815, 818, 871, 1194, 1336, 1472, 1491, 1499, and 1528.

## A.4 LIMITATIONS

SynSQL demonstrates strong performance in generating synthetic databases for text-to-SQL evaluation, but it has limitations. The schema selection process may omit relevant tables or columns, leading to gold queries returning empty results. This remains an active area of research in text-to-SQL evaluation. The challenge is amplified in our data synthesis setting, where the absence of actual database contents and value-based retrieval mechanisms makes high-recall schema selection inherently difficult. However, several practical extensions could improve robustness while maintaining the minimalist design principle. First, implementing multi-hop schema traversal guided by LLMs could recover essential join paths and connector tables in complex schemas, adding minimal columns while significantly boosting recall. Second, employing ensemble methods across multiple LLMs could reduce interpretation variance and yield more stable column predictions. These approaches offer promising directions for addressing the remaining failure cases while preserving SynSQL's core advantages.

Additionally, SynSQL relies on the assumptions made by the large language models used. If the LLMs misinterpret the question intent or generate inconsistent data, this can lead to lower success rates. Incorporating additional constraints or validation steps during data generation could help mitigate this.

## A.5 ANALYSIS ON EFFECT OF CRITIC COMPONENT

As shown in Figure 2, the critic component consistently improves SynSQL's success rate across different LLMs on the BIRD dev set. To provide deeper insight into this improvement, Figure 9 breaks down the critic's impact across its five evaluation criteria: hint alignment, key integrity, schema coverage, data complexity, data variety, and relevance. The critic consistently enhances performance across all dimensions, indicating its effectiveness in generating more semantically coherent and diverse synthetic databases.

**Critic's Role in Avoiding Oversimplified Data Patterns**   One potential limitation of LLM-based synthesis is the tendency to generate overly simplistic or repetitive data patterns, which could artificially inflate success rates without providing meaningful evaluation coverage. The critic component addresses this by explicitly evaluating data complexity and variety as core quality dimensions. This improvement is reflected in the agreement rate (AR), which quantifies how well the synthetic database's query evaluations align with those of the original human-curated database on a per-query basis. As shown in Figure 10, the critic significantly improves AR across all three LLMs tested on the BIRD dev set. This indicates that the critic component enhances not just semantic alignment with question intent, but also the fundamental ability to differentiate between correct and incorrect SQL queries, the core objective of robust evaluation databases.

**Feedbacks**   Through detailed analysis of critic feedback across our experimental runs, we observed consistent patterns in how the critic identifies and addresses data quality issues. The critic provides targeted feedback such as: figure 11 shows an example where the critic highlights deficiencies in data complexity and variety, prompting the synthesizer to regenerate synthetic data that better aligns with the question intent and enhances evaluation robustness.

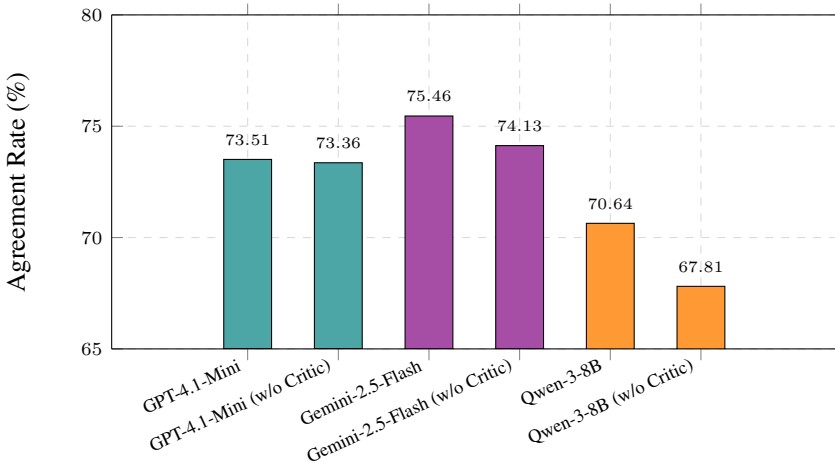

Figure 9: Impact of the critic component on each of the five data quality criteria in SynSQL, using Gemini-2.5-Flash on Spider and BIRD dev sets. Spider results exclude the Hint Alignment criterion as evidence/hint entries are not present in Spider.

Figure 10: Impact of the critic component on agreement rate (%) of SynSQL with three different LLMs on the BIRD dev set. Showing that the critic improves the database's ability to distinguish between correct and incorrect SQL queries.

**Question**: In the race a driver set the lap record for the Austrian Grand Prix Circuit, how long did he spend at the pit stop at that same race?

**Evidence**: lap record means the fastest time recorded which refers to time, how long spent at pitstop refers to duration

**Feedback**:
Add explicit marking or clearer linkage of the lap record in lapTimes, possibly by including multiple laps per driver and highlighting the fastest lap. Introduce edge cases such as multiple pit stops per driver and multiple drivers with close lap times to increase complexity. Increase variety by adding more races, circuits, and drivers to better simulate real-world data.

Figure 11: Example of critic feedback highlighting deficiencies in data complexity and variety, prompting regeneration of synthetic data to better align with question intent and evaluation robustness.

The critic also frequently identifies key integrity violations, such as non-unique primary keys or referential integrity breaches. Figure 12 illustrates an example where the critic detects foreign key violations, leading to regeneration that enforces these constraints and ensures schema integrity.

**Question**: What is the total amount of Community College District closure in 1989 in the city of San Francisco?

**Evidence**: N/A

**Feedback**:
Ensure all CDSCode values in "schools" exist in "frpm" and "satscores" to maintain foreign key integrity.
Increase complexity by including more diverse DOCType values and closure scenarios.
Add more variety in StatusType or closure reasons to enrich the dataset.

Figure 12: Example of critic feedback highlighting issues in foreign key integrity, leading to regeneration that enforces schema integrity.

More examples of critic feedback are shown in Figures 13 and 14, demonstrating the critic's consistent role in identifying and rectifying data quality problems.

**Question**: How many patients with a normal level of complement 3 have a P pattern observed in the sheet of ANA examination?

**Evidence**: normal level of complement 3 refers to C3 > 35; have a P pattern observed in the sheet of ANA examination refers to ANA Pattern = 'P'; Should compute the number of distinct ones

**Feedback**:
Adjust the 'C3' values in the 'Laboratory' table to better reflect the "normal level of complement 3 refers to C3 > 35" hint. Ensure a good mix of values both above and below 35, with a clear distinction for "normal" cases. For example, include more values significantly above 35 for "normal" cases, and values significantly below 35 for "abnormal" cases.

Figure 13: An example of critic feedback

> **Question**: Which country is the oldest driver from?
>
> **Evidence**: date of birth refers to drivers.dob; The larger the birthday value, the younger the person is, and vice versa
>
> **Feedback**:
> Increase the variety and range of birth dates to better highlight the oldest driver and include edge cases such as multiple drivers born on the same day or very close dates. Add explicit foreign key relationships or at least ensure driverId is consistently referenced across tables to improve schema integrity. Expand the drivers table with more entries and nationalities to increase variety and complexity. Although lapTimes and results are less relevant, adding meaningful data or relationships could improve overall schema coverage and complexity.

Figure 14: An example of critic feedback

This systematic feedback mechanism ensures that subsequent iterations produce more robust test databases that can effectively distinguish between semantically correct and incorrect SQL queries. Overall, the critic's feedback focuses on: (1) key integrity and schema coverage to ensure structural validity, (2) presence of edge cases and boundary values, (3) diversity in categorical attributes, (4) realistic distributions that reflect real-world data patterns while aligning with question intent, and (5) inclusion of potential query failure scenarios. This multi-dimensional assessment prevents the framework from converging on overly simplistic data that might mask SQL generation errors, ensuring that high success rates reflect genuine semantic alignment and increases the robustness of evaluation, rather than artificially accommodating weak queries.

## A.6 EVALUATION WITH STRONGER LANGUAGE MODELS

In our main experiments, we focused on evaluating SynSQL with three small size LLMs: GPT-4.1-Mini, Gemini-2.5-Flash, and Qwen-3-8B. To further assess SynSQL's performance, we conducted additional experiments using a more powerful model, GPT-4.1. The results, presented in Table 5, demonstrate that SynSQL continues to outperform the vanilla GPT-4.1 model across all metrics on the BIRD dev set.

Table 5: Performance comparison of SynSQL using GPT-4.1-Mini against GPT-4.1 on BIRD dev set.

| Method | Success Rate (SR) | Agreement Rate (AR) | |
|---|---|---|---|
| | | DAIL-SQL | DIN-SQL |
| BIRD (Original) | 99.87 | 100.00 | 100.00 |
| Vanilla GPT-4.1-Mini | 69.43 | 58.57 | 61.88 |
| SynSQL(GPT-4.1-Mini) | 82.07 | 73.51 | 78.90 |
| Vanilla GPT-4.1 | 76.09 | 66.52 | 70.58 |
| SynSQL(GPT-4.1) | **86.51** | **74.69** | **79.51** |

## A.7 EVALUATION ON SPIDER 2.0

We also evaluated SynSQL on the Spider 2.0 dataset (Lei et al., 2024), which comprises 632 real-world text-to-SQL workflow problems derived from enterprise-level database use cases. Since most of these problems are based on Snowflake and BigQuery databases, we focused on the 135 questions that can be executed on SQLite to ensure compatibility with our synthetic database generation framework. We then evaluated all questions that have gold SQL queries provided.

As shown in Table 6, SynSQL outperforms the vanilla LLM baseline on success rate demonstrating its effectiveness in generating synthetic databases that align with question intent even in more complex, real-world scenarios.

Table 6: Performance comparison of SynSQL on Spider 2.0 dataset.

| Method | Success Rate (SR) |
|---|---|
| Vanilla GPT-4.1-Mini | 54.17 |
| SynSQL(GPT-4.1-Mini) | **58.33** |

## A.8 EVALUATION ON ADVANCED TEXT-TO-SQL SYSTEMS

To further validate SynSQL's effectiveness, we evaluated its performance using three advanced text-to-SQL systems: Gemini-SQL (Multitask SFT + Gemini-2.5-Pro) (Pourreza & Kubik, 2025), which ranks at the top of the BIRD leaderboard in the single-model track at the time of writing, OmniSQL-32B (Li et al., 2025), and CSC-SQL-32B (Sheng & Xu, 2025). These models represent the state-of-the-art in text-to-SQL generation and provide a robust benchmark for assessing the quality of synthetic databases generated by SynSQL.

As shown in Table 7, SynSQL consistently outperforms vanilla LLM baselines across all three advanced text-to-SQL systems on the BIRD dev set and achieves substantial agreement rates based on Cohen's Kappa score range. Notably, SynSQL maintains the ranking of text-to-SQL models observed with the original human-curated databases, demonstrating its ability to preserve evaluation fidelity.

These results underscore SynSQL's versatility and effectiveness in generating high-quality synthetic databases that facilitate robust evaluation of text-to-SQL systems, even when leveraging cutting-edge models.

Table 7: Performance comparison of SynSQL and baseline methods on BIRD dev set using advanced text-to-SQL systems: Gemini-SQL, OmniSQL, and CSC-SQL. AR: Agreement Rate (%), EX: Execution Accuracy (%), $\Delta$: Difference in Execution Accuracy between original and synthetic databases.

| Method | Gemini-SQL | | | OmniSQL | | | CSC-SQL | | |
|---|---|---|---|---|---|---|---|---|---|
| | AR↑ | EX | $\Delta\downarrow$ | AR↑ | EX | $\Delta\downarrow$ | AR↑ | EX | $\Delta\downarrow$ |
| BIRD (Original) | 100.00 | 72.10 | - | 100.00 | 66.75 | - | 100.00 | 71.12 | - |
| Vanilla GPT-4.1-Mini | 48.67 | 84.68 | 12.58 | 47.72 | 80.51 | 13.76 | 47.70 | 82.20 | 11.08 |
| Vanilla Gemini-2.5-Flash | 56.82 | 82.86 | 10.76 | 55.71 | 78.42 | 11.67 | 55.11 | 80.70 | 9.58 |
| SynSQL(Qwen-3-8B) | 65.49 | 74.64 | 2.54 | 61.62 | 68.45 | 1.70 | 62.05 | 73.19 | 2.07 |
| SynSQL(Gemini-2.5-Flash) | **68.67** | 73.21 | 1.11 | **68.73** | 66.69 | **0.06** | **65.14** | 70.08 | **1.04** |
| SynSQL(GPT-4.1-Mini) | 65.58 | 73.14 | **1.04** | 65.61 | 65.71 | 1.04 | 63.98 | 68.12 | 3.00 |

## A.9 REALISM AND MINIMALISM OF SYNTHETIC DATABASES

SynSQL-generated databases are not only realistic but also minimal and lightweight. This property is particularly valuable in scenarios where gold queries are unavailable, not only during synthesis but also for evaluation. For example, in production or cold-start settings, it is crucial to inspect and validate the generated database, either through human-in-the-loop processes or by generating expected outputs via table reasoning. The compactness of SynSQL databases facilitates such inspection and validation, making them practical for robust evaluation even when large-scale or gold-standard annotations are not accessible.

We saw in Figure 4 that in SynSQL we have an average of 8.37 columns for BIRD and 6.71 columns for Spider to answer a query, significantly fewer than the full schemas of 75.56 and 24.55 columns respectively. We see an example of this in question 1000 from the formula_1 database (see Figure 15), SynSQL generates a minimal database with only 2 tables and 9 columns, compared to the original database's 13 tables and 94 columns. The synthetic database sufficiently covers the question and relevant edge cases while being just 20KB in size, whereas the original is 21,836KB, making SynSQL's output much easier to inspect and validate. In contrast, synthetic databases generated by prior work such as TestSuiteAccuracy (TSA) often contain random values from fuzzing and are

typically as large as the original databases. As illustrated in Figures 16 and 17, which show the entirety of data generated for this question by SynSQL, the synthetic data includes realistic values that closely match the question intent.

> **Question**: Which racetrack hosted the most recent race? Indicate the full location
>
> **Evidence**: full location refers to location+country; most recent race = MAX(date)
>
> **Gold Query**: SELECT T1.location FROM circuits AS T1 INNER JOIN races AS T2 ON T1.circuitId = T2.circuitId ORDER BY T2.date DESC LIMIT 1

Figure 15: An example from the `formula_1` database (question 1000).

Moreover, SynSQL ensures that values within each row are meaningfully related and contextually accurate. For example, if a row in the `races` table has the year set to `2024`, all corresponding data in that row (such as race name or date) is consistent with that year. Similarly, in the `circuits` table, if the location is `Monza`, the country is set to `Italy`, reflecting the real-world fact that there is a Formula 1 Grand Prix held in Monza, Italy. This level of realism and consistency, both within rows and across related tables, is achieved by leveraging LLMs to generate data that maintains semantic coherence and factual alignment.

| raceID | year | circuitID | name | date |
|--------|------|-----------|------|------|
| 101 | 2022 | 1 | British Grand Prix | 2022-07-03 |
| 102 | 2022 | 2 | Monaco Grand Prix | 2022-05-29 |
| 103 | 2022 | 3 | Japanese Grand Prix | 2022-10-09 |
| 104 | 2022 | 4 | United States Grand Prix | 2022-10-23 |
| 105 | 2022 | 5 | Italian Grand Prix | 2022-09-11 |
| 106 | 2023 | 1 | British Grand Prix | 2023-07-09 |
| 107 | 2023 | 2 | Monaco Grand Prix | 2023-05-28 |
| 108 | 2023 | 3 | Japanese Grand Prix | 2023-10-08 |
| 109 | 2023 | 4 | United States Grand Prix | 2023-10-22 |
| 110 | 2023 | 5 | Italian Grand Prix | 2023-09-10 |
| 111 | 2024 | 1 | British Grand Prix | 2024-07-07 |
| 112 | 2024 | 2 | Monaco Grand Prix | 2024-05-26 |
| 113 | 2024 | 3 | Japanese Grand Prix | 2024-10-13 |
| 114 | 2024 | 4 | United States Grand Prix | 2024-10-27 |
| 115 | 2024 | 5 | Italian Grand Prix | 2024-09-08 |

Figure 16: Generated synthetic table `races` for question 1000 from the `formula_1` database. The synthetic data contains realistic values that align with the question intent.

| circuitID | name | location | country |
|-----------|------|----------|---------|
| 1 | Silverstone Circuit | Silverstone | United Kingdom |
| 2 | Circuit de Monaco | Monte Carlo | Monaco |
| 3 | Suzuka Circuit | Suzuka | Japan |
| 4 | Circuit of the Americas | Austin | USA |
| 5 | Autodromo Nazionale Monza | Monza | Italy |

Figure 17: Generated synthetic table `circuits` for question 1000 from the `formula_1` database. The synthetic data contains realistic values that align with the question intent.

## A.10 INCONSISTENCY EXAMPLES FROM SPIDER DEV SET

There are questions in the spider dev set that the gold query does not align with the content of original test databases. Below are some examples of such inconsistencies, which lead to the observed low success rates for the original Spider databases. SynSQL generates synthetic data that aligns with the

question intent and recovers such inconsistencies. For example, in (Figure 18) the questions asks for the location and name for all stadiums with a capacity between 5000 and 10000. However, there are no such stadiums in the original database, leading to the gold query returning empty results. SynSQL generates synthetic data that includes stadiums within this capacity range. Another example is shown in (Figure 19), where the question asks for the city and country of the Alton airport. However, the original database `flight_2` has the airport name listed as `Alton`, with a trailing space, leading to a mismatch with the gold query. SynSQL generates synthetic data that correctly matches the airport name as specified in the question.

> **Question**: Show location and name for all stadiums with a capacity between 5000 and 10000
>
> **Evidence**: N/A
>
> **Gold Query**: SELECT LOCATION , name FROM stadium WHERE capacity BETWEEN 5000 AND 10000

Figure 18: An example of inconsistencies between gold query and database contents in the Spider dev set. SynSQL aligns with the question, leading to recovery of such inconsistencies.

> **Question**: Which city and country is the Alton airport at?
>
> **Evidence**: N/A
>
> **Gold Query**: SELECT City, Country FROM AIRPORTS WHERE AirportName = "Alton"

Figure 19: An example of inconsistencies between gold query and database contents in the Spider dev set. SynSQL aligns with the question, leading to recovery of such inconsistencies.

## A.11    COLUMN SELECTION PROMPT

```
You are an expert data analyst. Your task is to carefully review the database
schema, understand the question, and use the hint to determine which columns
from which tables must be populated with synthetic data to fully support
answering the question.

This task is for synthetic data generation, NOT for Text2SQL. In this context,
RECALL IS MORE IMPORTANT THAN PRECISION. It is better to include more columns
than to miss important ones.

Database Schema:
{DATABASE_SCHEMA}

This schema defines the database structure, including tables, columns, primary
keys, foreign keys, and relevant relationships or constraints.
You can also rely on the following descriptions for the columns to better
understand the nature of data that would be generated for them.

Column Descriptions:
{COLUMNS_DESCRIPTIONS}

Question:
{QUESTION}

Hint:
{HINT}

The hint is intended to guide your attention to the specific elements of the
database schema that are essential for addressing the question accurately

Task:
Based on the database schema, question, and hint provided, your task is to
determine the columns from tables that need to be populated with data to support
the question.
You should also provide the foreign keys that are needed to potentially join the
tables, in the context of the question.
For each of the selected columns, explain why exactly it is necessary to
generate data for, in order to cover the question. Your explanation should be
logical and concise, demonstrating a clear understanding of the database schema,
the question, and the hint.

Please respond with a JSON object structured as follows:

```json
{{
  "chain_of_thought_reasoning": "Your reasoning for selecting the columns, be
concise and clear.",
  "table_name1": ["column1", "column2", ...],
  "table_name2": ["column1", "column2", ...],
  ...
  "foreign_keys": ["table_name1.column1 = table_name2.column2, ...]
}}
```

Make sure your response includes the table names as keys, each associated with a
list of column names that are necessary for generating synthetic data that would
be enough to support the question.
For foreign keys, make sure you include foreign keys within tables that are
needed to cover the possibility of join, IN CONTEXT OF THE QUESTION AND THE
HINT.
For each aspect of the question, provide a clear and concise explanation of your
reasoning behind selecting the columns. Only output a json as your response.
```

Figure 20: The prompt template used for column selection in the schema selector component of SynSQL.

## A.12 COLUMN EXPANSION PROMPT

```
You are an expert data analyst. Your task is to analyze the provided database
schema and a list of already selected columns, and identify the most
semantically similar columns to the selected ones.

Database Schema:
{DATABASE_SCHEMA}

This schema defines the database structure, including tables, columns, primary
keys, foreign keys, and relevant relationships or constraints.
You can also rely on the following descriptions for the columns to better
understand the nature of the data that would be generated for them.

Column Descriptions:
{COLUMNS_DESCRIPTIONS}

Already Selected Columns:
{SELECTED_COLUMNS}

Task:
Based on the database schema, column descriptions, and the already selected
columns, your task is to identify, AT MOST 3 of the most semantically similar
columns, that are:
1. Semantically similar to the selected columns but in a different table (e.g.
if Country.id is selected, then Match.country_id would be a similar column)
OR
2. Likely to contain data that would complement the selected columns

Please respond with a JSON object structured as follows:

```json
{{
  "chain_of_thought_reasoning": "Your reasoning for selecting additional
columns, be concise and clear.",
  "table_name1": ["additional_column1", "additional_column2"],
  "table_name2": ["additional_column1"],
}}
```

Make sure your response includes ONLY NEW columns that weren't in the original
selection. Do not repeat columns that were already selected.
Your response should only include 3 columns in total (for all tables), NO MORE.
So pick the most important ones.
For each additional column, briefly explain why you think it's similar or
related to the already selected columns.

Only output a json as your response.
```

Figure 21: The prompt template used for column expansion in the schema selector component of SynSQL.

## A.13 DATA SYNTHESIZER PROMPT

```
You are an expert data generator for SQL databases. Your task is to create
realistic and challenging test data that will properly test a system's ability
to answer complex questions.

For the given question, use the schema and hint to generate SQLite test data for
the database. The schema identifies the specific tables and columns that are
relevant to the question, and the hint provides guidance on how to structure the
data to make the question answerable.

Use the following instructions for generating the test data:
1- Pay attention to the primary key and foreign key constraints to ensure data
integrity.
2- Make sure the data includes edge cases and is challenging to answer the
question.
3- Include a variety of data that covers different scenarios related to the
question.
4- Generate enough data to make the question answerable but also challenging.
5- The data should be realistic and diverse.
6- Your response should follow the EXACT format of the example, where every line
starts with INSERT. DO NOT group the insert statements and DO NOT put values on
a different line than the INSERT statement.
7- IMPORTANT: For each INSERT statement, ensure the number of values EXACTLY
matches the number of columns in the table. Count the columns carefully in the
CREATE TABLE statement and provide exactly that many values in each INSERT
statement.

{FEEDBACK_INSTRUCTION}

{ONE_EXAMPLE}

Schema of the database with question and hint:

Database: {DB_NAME}

Schema: {DATABASE_SCHEMA}

Question: {QUESTION}

Hint: {HINT}
```

Figure 22: The prompt template used for data synthesis component of SynSQL.

## A.14 DATA CRITIC PROMPT

```
You are a data critic agent designed to evaluate synthetic data for answering
natural language questions. Your task is to analyze the generated data and
determine if it is correct, sufficient, complex, and diverse enough to answer
the question.

You should evaluate the data based on the following criteria:
1. Hint Alignment: Does the data follow the intent and details of the question
hint?
2. Key Integrity: Does the data respect uniqueness and foreign key relationships
in the schema?
3. Schema Coverage: Does the data include the relevant columns and relationships
from the schema?
4. Complexity: Does the data include sufficient complexity and edge cases?
5. Variety: Is there enough variety in the data?
6. Relevance: Is the data directly related to answering the question?

{ONE_EXAMPLE}

Question: {QUESTION}

Database Schema: {DATABASE_SCHEMA}

Hint: {HINT}

Generated Data: {GENERATED_DATA}

Provide a detailed evaluation of the data based on the criteria of Hint
Alignment, Key Integrity, Schema Coverage, Complexity, Variety, and Relevance.
For each criterion, provide a score from 1-10 and specific feedback on what
aspects need improvement.
If there are issues such as incorrect data, violations of key integrity (e.g.,
non-unique or missing foreign keys), or other schema-related errors, provide
clear and actionable feedback to help address and resolve these problems.
When providing feedback, consider that having more data is usually more
beneficial, provided it does not distract from or obscure the key information
required to answer the question. Try not to recommend reducing the data.
Finally, determine if the data meets the minimum quality criteria to answer the
question effectively.
```

Figure 23: The prompt template used for data critic component of SynSQL.

## A.15 THE USE OF LARGE LANGUAGE MODELS (LLMS)

In accordance with the ICLR 2026 Code of Ethics, we acknowledge that large language models were used to assist with the polishing of the writing in this paper.

