# OpenReview forum: "SynSQL: Synthetic Database Generation for Robust Evaluation of Text-to-SQL Systems"
_ICLR.cc/2026/Conference — Submitted to ICLR 2026_

### Official Review · Reviewer_wLfc · 2025-10-29

**Soundness:** 2
**Presentation:** 2
**Contribution:** 2
**Rating:** 4
**Confidence:** 3

**Summary:**

The paper proposes a method to automatically generate synthetic databases for text to SQL evaluation using only the natural language question and the schema. The approach enforces schema constraints such as primary and foreign keys while generating realistic data. It uses a three step approach with schema selection, data generation, and a critic that applies feedback recursively up to three times. The synthetic datasets generated by GPT-4.1-mini, Gemini-2.5-Flash, and Qwen-3-8B are then used to evaluate text to SQL models DIN-SQL and DAIL-SQL and are compared against human curated datasets like Spider and BIRD. The authors argue that this method reduces dependency on manually curated data and enables evaluation in privacy sensitive or cold start scenarios. The experiments show that the generated data shows alignment with human curated datasets and can respect schema constraints better than human curated datasets in some cases.

**Strengths:**

- The approach targets cold start and privacy preserving settings where only the schema and the question are available.

- By enforcing schema constraints during generation, the method reduces errors common in human curated datasets and addresses a valid concern in benchmark design.

- The paper identifies weaknesses in existing text to SQL datasets like inconsistencies (extra space, lower/upper case).

**Weaknesses:**

- The evaluation uses DIN-SQL and DAIL-SQL and older dataset versions, while newer versions such as Spider 2 and BIRD interact are not used, which limits relevance given the rapid progress in LLMs.

- The data generators are weaker models, and stronger models might need less engineering to generate a synthetic dataset of good quality. While using a cheap model is desirable, the cost of generating an evaluation dataset with a strong model is lower because the dataset is generated infrequently.

- The experimental design varies three weak generators and evaluates only two older systems. A stronger design would use a strong generator to produce a dataset  and then benchmark multiple systems on it.

- The content and construction of the auxiliary knowledge K are not described, although the code and the appendix show it is evidence on BIRD and empty on SPIDER. This should be specified in the main text.

- The post processing step may change semantics due to padding with NULLs or truncation, and its effects on the evaluation are unclear.

- Main results lack details about vanilla baselines and how they are used to generate datasets.

- The effects of schema reduction are insufficiently discussed, since extra columns might increase difficulty for model evaluation, and reducing to a subset may cause queries that work on the original dataset to fail (in other words they use the extra columns but get the correct output).

- The prompts for the critic and the synthesizer include the placeholder {ONE_EXAMPLE}, and the source of this example is not specified.

- The data validation step prior to the data critic might change the generated data, and it seems the results of this step are not provided back as feedback.

**Questions:**

- The column expansion prompt enforces a limit of 3 columns. Is this step applied once or recursively?

- When using DIN-SQL and DAIL-SQL, is the model GPT-4 as in the original work?

- When generating inserts, if the number of values does not match the number of columns, is there a reason not to use structured output to enforce these constraints?

- For the data critic, what was the reason to use a 1 to 10 score instead of a simple pass or fail for each criterion?

- Minor typo, in line 451, 2 should be Figure 2.

---

> ### Author Response · Authors · 2025-11-21
> **Authors' response to reviewer**
>
> We sincerely thank the reviewer for their valuable feedback and comments on our paper.
>
> > **W1: Use of older systems**
>
> Regarding the choice of DIN-SQL and DAIL-SQL, our goal is to evaluate test-database discriminativeness, not advance text-to-SQL modeling. DIN-SQL and DAIL-SQL remain two of the most prominent publicly available systems whose predictions are reproducible and runnable at scale. Other systems at the top of the leaderboard provide results only through submissions to the Spider or BIRD organizers and do not release their query outputs or implementation details, preventing execution on our synthetic data. DIN-SQL and DAIL-SQL also differ significantly in reasoning style (decomposition vs. few-shot selection), making them ideal for studying whether SynSQL preserves relative model ranking.
>
> > **W2 & W3:  Would strong generators trivialize the problem? Newer benchmarks?**
>
> To address your concern and further strengthen our evaluation, we also report results using a stronger **GPT-4.1** model and run additional experiments on the **Spider 2.0** dataset. The results using GPT-4.1, presented in the new Table 5, demonstrate that SynSQL continues to outperform the vanilla GPT-4.1 model across all metrics on the BIRD dev set.
>
> Spider 2.0 is largely based on Snowflake and BigQuery, which are different from the SQLite schemas that SynSQL is based on. Nevertheless, we generate data for all SQLite-subset questions in Spider 2.0-lite, demonstrating that SynSQL achieves a 58.33% success rate, compared with 54.17% for the vanilla version.
>
> These results, shown in the new Sections A.6 and A.7, confirm that SynSQL’s improvements hold across different models and benchmark databases.
>
> > **W4: Auxiliary knowledge `K` not sufficiently described**
>
> We clarify that auxiliary knowledge consists of evidence/hints in BIRD (as now described in Section 4.1 of the updated paper, highlighted) and is empty for Spider. We have added this specification to the main text for clarity.
>
> > **W5: Post-processing may change semantics**
>
> The post-processing step, specifically the NULL padding and truncation, never occurs in SynSQL because the reduced schema space achieved through schema selection prevents the issue entirely. In the vanilla version, we ensure that padding and truncation occur only at the attribute level, so they do not affect the overall semantics.
>
> > **W6: Vanilla baselines not fully described**
>
> Following your suggestion, we added clarifications about the details of the vanilla baseline in Section 5.1 of the updated paper. For our vanilla baseline, we prompt the LLM to generate synthetic data in a single pass without schema reduction or critic feedback. The vanilla baseline is equivalent to SynSQL with only the synthesizer component, using the same prompting strategy but operating on the full schema without iterative refinement.
>
> > **W7: Effects of schema reduction**
>
> Sec. 5.3 now includes explicit discussion: extra columns can inflate difficulty, but aggressive reduction risks omitting required columns. The ablation table quantifies this trade-off and shows that our ensemble-expansion strategy yields the best balance.
>
> > **W8: Source of `{ONE_EXAMPLE}` placeholder**
>
> The `{ONE_EXAMPLE}` placeholder in prompts is a domain-specific example, which you can view in the submitted anonymous codebase linked below:
>
> [https://github.com/author-23179/ICLR-23179/blob/main/src/prompts/BIRD/system_test_generation.txt](https://github.com/author-23179/ICLR-23179/blob/main/src/prompts/BIRD/system_test_generation.txt)
>
> > **W9: Data validation not fed back to critic**
>
> The data validation step happens after the synthesis and prior to the process of the critic. So the feedbacks are based on the formatted and already validated data.
>
> ---
>
> ## Questions:
> > Q1: The column expansion prompt enforces a limit of 3 columns. Is this step applied once or recursively?
>
> This step is applied only once to prevent excessive expansion of the schema space
>
> > Q2: When using DIN-SQL and DAIL-SQL, is the model GPT-4 as in the original work?
>
> We have used the query results from the original work.
>
> > Q3: is there a reason not to use structured output to enforce these constraints?
>
> In our preliminary results, when the schema becomes large enough the models struggles to maintain schema integrity regardless of the output construction format.
>
> > Q4: For the data critic, what was the reason to use a 1 to 10 score instead of a simple pass or fail for each criterion?
>
> The 1-10 scale provides granular feedback for iterative improvement. We also wanted to measure the exact effectiveness of feedbacks on each criterion which you can see in the new Figure 9.
>
> > Q5: Minor typo
>
> Typo has been corrected in revision.
>
> ---
>
> We hope the detailed answers and additional results address your concerns. We kindly ask you to consider the possibility of a score adjustment.

---

> > ### Comment · Reviewer_wLfc · 2025-11-26
> >
> > I thank the authors for their detailed response and new experiments.
> > I still have some concerns.
> >
> > - Regarding the statement that "submissions to the Spider or BIRD organizers do not release their query outputs or implementation details" I would like to note that around thirty entries on the BIRD leaderboard link to public GitHub repositories and outperform both methods used in the paper.
> > - The results on Spider 2.0 show that the magnitude of improvement decreases from double digits on Spider to about 4% which suggests on newer benchmarks, the relative advantage of the proposed method becomes substantially smaller, and it is unclear how robust the gains will be as benchmarks evolve.
> > - For W9, I am not sure if I follow the authors' response. If the validator's changes are provided to the generator (not critic), this may primarily help the generator produce more valid data.
> > - For Q3, the structured output format in principle guarantees that the generated response will follow a specific format. For a specific table, the AST corresponding to INSERT statements can be represented using a JSON schema.

---

> > > ### Author Response · Authors · 2025-11-28
> > > **Authors' response to reviewer**
> > >
> > > Thank you once again for your valuable feedback and suggestions.
> > >
> > > > **More Advanced Text-to-SQL Systems**
> > >
> > > To address this concern and further strengthen our evaluation, we additionally report results using three advanced text-to-SQL models from the BIRD leaderboard: Gemini-SQL (Multitask SFT + Gemini-2.5-Pro), currently top of the leaderboard in the single-model track, OmniSQL-32B (Li et al., 2025), and CSC-SQL-32B.
> > >
> > > | Method | **Gemini-SQL** | | | **OmniSQL** | | | **CSC-SQL** | | |
> > > |--------|----------------:|----------------:|----------------:|----------------:|----------------:|----------------:|----------------:|----------------:|----------------:|
> > > |        | AR↑ | EX | Δ↓ | AR↑ | EX | Δ↓ | AR↑ | EX | Δ↓ |
> > > | **BIRD (Original)** | 100.00 | 72.10 | – | 100.00 | 66.75 | – | 100.00 | 71.12 | – |
> > > | **Vanilla GPT-4.1-Mini** | 48.67 | 84.68 | 12.58 | 47.72 | 80.51 | 13.76 | 47.70 | 82.20 | 11.08 |
> > > | **Vanilla Gemini-2.5-Flash** | 56.82 | 82.86 | 10.76 | 55.71 | 78.42 | 11.67 | 55.11 | 80.70 | 9.58 |
> > > | **SynSQL (Qwen-3-8B)** | 65.49 | 74.64 | 2.54 | 61.62 | 68.45 | 1.70 | 62.05 | 73.19 | 2.07 |
> > > | **SynSQL (Gemini-2.5-Flash)** | **68.67** | 73.21 | 1.11 | **68.73** | 66.69 | **0.06** | **65.14** | 70.08 | **1.04** |
> > > | **SynSQL (GPT-4.1-Mini)** | 65.58 | 73.14 | **1.04** | 65.61 | 65.71 | 1.04 | 63.98 | 68.12 | 3.00 |
> > >
> > >
> > > As shown in the above table and the new section A.8 of the revised paper, these results confirm that SynSQL’s improvements persist across different text-to-SQL architectures. The approach continues to yield substantial agreement rates based on Cohen’s kappa score range, and preserves the relative ranking observed on the original human-curated databases. This demonstrates that SynSQL maintains evaluation fidelity even when used with state-of-the-art models.
> > >
> > > > **Spider 2.0 Gains**
> > >
> > > Spider 2.0-lite is an exceptionally challenging benchmark,  even cutting-edge text-to-SQL models struggle on it, with the highest reported execution accuracy being only 55.21%. In that context, we believe a 4-percentage-point gain in our task is meaningful rather than marginal. Such an improvement reflects a nontrivial step forward in model robustness and generalization, especially as benchmarks grow harder.
> > >
> > > > **Data Validator**
> > >
> > > The validator performs purely syntactic corrections. These edits do not provide signals useful to the generator, as they address basic, rule-based errors, such as mismatches between the casing of database values and the casing specified in the question or hint. These are errors that we can catch every time. In contrast, the critic delivers feedback that drives semantic and schema-level refinement of the generated data to better align with the question intent and schema constraints.
> > >
> > > > **Output Format**
> > >
> > > We experimented with structured formats, but for large and complex schemas (especially in BIRD), the dominant failure mode was not formatting but the model’s difficulty maintaining relational consistency (e.g., foreign keys, multi-table coherence, realistic value distributions), as shown in Appendix A.3. Although structured formats enforce syntactic shape, they did not reduce these schematical and semantic errors and frequently lowered success rates by leading the model to generate overly simple data with fewer records. In contrast, generating SQL INSERT statements directly better evaluates the model’s ability to integrate the natural language question with the DDL schema and produce data consistent with both the schema and the text-to-SQL task.

---

### Official Review · Reviewer_wjhd · 2025-11-01

**Soundness:** 2
**Presentation:** 3
**Contribution:** 3
**Rating:** 4
**Confidence:** 4

**Summary:**

The paper introduced SynSQL, a framework for generating test databases for text-to-SQL tasks by synthesizing tables directly from natural language questions and schema structure, rather than relying on gold queries. Experiments on Spider and BIRD benchmarks show that SynSQL outperforms prior automated methods.

**Strengths:**

- The paper offers a fresh perspective on SQL data sythesis, which is more general and requires less input data.
- The synthetic dataset shows on-par or even better results with human curated dataset.
- Fully automated, easy to scale

**Weaknesses:**

- Baseline is very naive and not well established. Many comparsions are done against vanilla GPT4.1-mini and the gains are not surprising. Also can the authors elaborate more on the choice of DIN-SQL and DAIL-SQL vs the rest?
- Enforcing case sensitivity doesn't correctly reflect the real-world cases. The synthetic dataset / query can be much cleaner than the real ones.

**Questions:**

- When generating the relevant schema, why use different temperatures instead of multiple-sampling with a higher temperature or other more common sampling approach?
- The synthesizer does not rely on reference queries, then how to make sure the result is differentiable and minimal?
- Discussion or comparison with methods that generate synthetic natural language or SQL queries will be a lot interesting, since these two line of approaches are directly related.

---

> ### Author Response · Authors · 2025-11-21
> **Authors' response to reviewer**
>
> We sincerely thank the reviewer for their valuable feedback and comments on our paper.
>
> > **W1: Baseline strength and model choices (DIN-SQL, DAIL-SQL)**
>
> We agree that the baseline should be clearly motivated. Test Suite Accuracy (TSA) is the only public system that automatically synthesizes relational test data without relying on curated datasets, like XDATA and others do. Even then, TSA fails to execute successfully on 39% of BIRD questions. That is why we look at the original BIRD and Spider databases as strong human-curated baselines for our work.
>
> To address your concern and further strengthen our evaluation, we also report results using a stronger **GPT-4.1** model and run additional experiments on the **Spider 2.0** dataset. These results, shown in the new Sections A.6 and A.7, confirm that SynSQL’s improvements hold across different models and benchmark databases. These comparisons highlight the novelty and practical value of SynSQL: to our knowledge, it is the first framework capable of generating high-coverage test databases directly from natural language and schema structure at scale.
>
> Regarding the choice of DIN-SQL and DAIL-SQL, our goal is to evaluate test-database discriminativeness, not advance text-to-SQL modeling. DIN-SQL and DAIL-SQL remain two of the most prominent publicly available systems whose predictions are reproducible and runnable at scale. Other systems at the top of the leaderboard provide results only through submissions to the Spider or BIRD organizers and do not release their query outputs or implementation details, preventing execution on our synthetic data. DIN-SQL and DAIL-SQL also differ significantly in reasoning style (decomposition vs. few-shot selection), making them ideal for studying whether SynSQL preserves relative model ranking.
>
> > **W2: Case sensitivity not reflecting real-world scenarios**
>
> We agree that real-world databases often contain inconsistent casing and imperfect values. Our goal was not to enforce artificial cleanliness, but rather to remain consistent with the annotation conventions present in benchmarks such as BIRD. When the natural language question indicates a case-sensitive value (e.g., proper nouns, acronyms, entity labels), SynSQL preserves the original casing to ensure that both the predicted and gold queries are evaluated against the intended value of the question. Synthetic data that mismatches case introduces false negatives that do not stem from model errors.
>
> ---
>
> ## Questions:
>
> > Q1:
>
> State-of-the-art Text-to-SQL systems have access to database contents, which enables sampling approaches, but when generating databases, we only have the schema and the question as our input, so the only variable that we can modify is the temperature of the LLMs. Using fixed high-temperature sampling leads to unstable schema predictions and often loses anchor columns entirely. Varying temperatures across runs is more effective because:
>
> - Low temperatures capture high-precision anchors (columns strongly implied by the question).
> - Medium temperatures explore semantically adjacent columns.
> - High temperatures surface rare or connector columns needed for join-path completion.
>
> This multi-temperature ensemble produces higher recall with much lower variance than sampling repeatedly at a single high temperature.
>
> > Q2:
>
> SynSQL enforces minimality and differentiability through a combination of:
>
> Schema reduction: The schema selector drastically shrinks the candidate schema space, ensuring results remain focused on the minimal set of attributes required for the question. The column count in Table 4 of the paper shows that we are substantially closer to the oracle schema (the absolute minimal set necessary) than the full schema, confirming the minimality of the synthetic databases.
>
> Critic guidance: The critic explicitly flags: unused columns, schema integrity violations, data of low complexity, or data that do not correspond to question semantics. When these occur, a refinement loop re-synthesizes to ensure higher differentiability and schema integrity. We added a new Section A.5 to the paper to show this in detail.
>
> Value grounding in both schema and question semantics: Because values are generated directly from the schema and the natural-language question, the synthesizer naturally avoids irrelevant columns or relationships.
>
> Finally, we evaluate SynSQL using two well-established Text-to-SQL systems to show the differentiability of the generated data. The results reach “substantial agreement” with human-curated benchmarks on Cohen’s kappa scale, confirming that SynSQL’s databases reliably distinguish between correct and incorrect queries.
>
> > Q3:
>
> Following your suggestion, we have updated the related work section to include this discussion.
>
> ---
>
> We hope the detailed answers and updated sections in the paper address your concerns. We kindly ask you to consider the possibility of a score adjustment

---

### Official Review · Reviewer_FQEo · 2025-11-01

**Soundness:** 3
**Presentation:** 3
**Contribution:** 3
**Rating:** 4
**Confidence:** 3

**Summary:**

The paper introduces a framework, SynSQL, to synthesize test databases by using LLMs for question understanding and aligning semantics with schema constraints. This framework has three stages: a schema selector, synthesizer, and critic.  The schema selector identifies relevant schema components and reduces the schema space. The synthesizer generates the test data based on the natural language question and reduced schema. Then, the critic reviews the quality of generated data and provides feedback for improvement. The resulting synthesized data complements the coverage of human-curated benchmarks and achieves higher success rate and agreement rate compared to human-authored datasets and past test generation methods respectively.

**Strengths:**

S1. The problem of generating a good-coverage test data is well motivated.
S2. The design of SynSQL is reasonable, especially the inclusion of Critic.

**Weaknesses:**

W1. If the framework's performance is generally comparable to a human-generated benchmark, the contribution might be considered incremental. To enhance the framework's usefulness, the resulting benchmark should create more accurate and challenging scenarios that can better expose the limitations of current methods.

W2. Although the framework is well-designed, there is a risk that LLMs may fail to diversify and instead repeat on a narrow range of prompt types. This could lead to the issue mentioned in lines 217-219 (about "too many WHERE operations" or other operations outside the scope), but instead this behavior can result in higher success rate and not as much prompt diversity.

**Questions:**

Q1. What kinds of critic does the model normally give? The prompt in A.9 includes "sufficient complexity". How does the model usually interpret that? I ask this to understand how likely W2 might occur and whether the model is able to identify such a situation.

---

> ### Author Response · Authors · 2025-11-21
> **Authors' response to reviewer**
>
> We appreciate the reviewer’s thoughtful feedback and recognition of the motivation, design, and utility of SynSQL.
>
> > **W1: Incrementality and benchmark usefulness**
>
> We appreciate the concern regarding the contribution relative to human-curated benchmarks. Section 5.2 and Appendix A.9, now more clearly articulate that SynSQL does not merely match human-generated data; it enhances their coverage in ways that humans systematically miss. SynSQL is built to uncover errors in human-built benchmarks as it exposes issues like mismatched value conventions (for example, casing differences between the question and the database), foreign-key problems, and cases where the question’s meaning doesn’t line up with the database content. As a result, we are able to demonstrate fixes for inconsistencies in Spider and identify evaluation errors in BIRD, helping produce databases that support more accurate evaluation. SynSQL also makes it possible to evaluate models in domain-specific, cold-start, or privacy-sensitive settings where real data is unavailable or restricted, something human-constructed benchmarks can’t provide.
>
> > **W2: Risk of LLMs repeating narrow prompt patterns**
>
> This is a very good comment and accordingly we have added an entire new Section A.5 to the appendix with a detailed analysis on this topic. As you noted, a potential limitation of LLM-based synthesis is the tendency to generate simplistic or repetitive data patterns, which could have high success rates without providing meaningful evaluation coverage. The critic component is precisely designed to address this by explicitly evaluating data complexity and variety as core quality dimensions.
>
> The impact of the critic shows up not just in higher success rates, but also in higher agreement rates, which measures how well the synthetic database distinguishes between correct and incorrect model predictions.
> In the new Figure 10, we show that the critic boosts agreement rates for all three LLMs on the BIRD dev set, reaching the “substantial agreement” range on Cohen’s kappa scale. This means the critic helps avoid the usual pitfalls of LLM-generated data, lack of diversity and insufficient complexity, and improves the system’s ability to tell good SQL predictions from bad ones, which is the whole point of a robust evaluation database.
>
>
> Additionally, the new Figure 9 further shows that the critic’s own evaluation metrics consistently improve after synthesizer incorporates its feedback, meaning the critic recognizes that the revised data is indeed more solid, diverse, and complex.
>
> ---
>
> ## Questions:
>
> >  What kinds of critic does the model normally give? The prompt in A.9 includes "sufficient complexity". How does the model usually interpret that?
>
> In the same section A.5, we’ve added several examples of critic feedback, shown in Figures 11–14. By analyzing feedbacks across our experimental runs, we noticed clear, recurring patterns in how the critic flags and fixes data quality issues. When it comes to ensuring “sufficient complexity,” the critic tends to focus on four things:
>  1. whether edge cases and boundary values are included
>  2. diversity in categorical attributes
>  3. realistic data distributions that match real-world patterns and stay aligned with the question intent
>  4. the presence of scenarios where queries could reasonably fail
>
> This kind of multi-dimensional checking keeps the system from falling into the trap of generating simple, uniform data that might hide SQL generation errors. As a result, high success rates actually reflect meaningful semantic alignment and stronger evaluation robustness, rather than data that conveniently accommodates weaker queries.
>
> ---
>
> We hope the detailed answers and updated sections in the paper address your concerns. We kindly ask you to consider the possibility of a score adjustment.

---

### Official Review · Reviewer_vdvq · 2025-11-03

**Soundness:** 2
**Presentation:** 3
**Contribution:** 2
**Rating:** 6
**Confidence:** 3

**Summary:**

The paper presents SynSQL, a novel framework designed to create synthetic evaluation test databases for Text-to-SQL systems. The approach assumes input consisting of a natural language (NL) question, a database schema, and associated knowledge. The framework features a modular design: a Schema Selector to focus on relevant columns; a Synthesizer that leverages Large Language Models (LLMs) to populate tables based on question semantics; and a Critic with an iterative refinement loop to ensure data integrity and quality. Experiments conducted on the widely used Spider and BIRD benchmarks successfully demonstrate that SynSQL produces effective test data that complements human-curated datasets. Furthermore, the framework notably outperforms a prior automated baseline, validating its utility for robust Text-to-SQL evaluation.

**Strengths:**

- The paper addresses a central, challenge in Text-to-SQL evaluation: the static nature of current benchmarks.
- The framework successfully generates robust test databases without relying on the gold SQL query, which is a significant advancement over prior work and enables broader applicability in cold-start or privacy-sensitive scenarios.

**Weaknesses:**

- The paper acknowledges that the Schema Selector may sometimes omit relevant tables or columns leading to gold queries returning empty results. This indicates that the recall-prioritizing ensemble expansion is not fully robust, especially when dealing with ambiguous schema mappings.
- The framework’s reliance on LLMs for both data synthesis and the critic feedback loop makes it sensitive to: Misinterpretation of Question Intent: 32.1% of failures were attributed to SynSQL's misalignment with the expected gold query, such as case-sensitivity mismatches or generating overly specific values.

**Questions:**

The ablation study highlights that the Schema Selector is critical, yet the error analysis shows that Schema Selection Omission accounts for nearly half of the failure cases (47.6%). Could the authors elaborate on the qualitative difference between selection failures caused by the "w/o Expansion" variant and the full SynSQL variant? Furthermore, how can the framework be practically extended to ensure higher recall on complex, sparsely connected schemas without dramatically increasing the column count and sacrificing minimalism?

---

> ### Author Response · Authors · 2025-11-21
> **Authors' response to reviewer**
>
> We sincerely thank the reviewer for their valuable feedback and comments on our paper.
>
>
> > **W1: Schema Selector Omissions**
>
> We agree that schema selection is critical (in SynSQL and many other leading text-to-SQL models). SynSQL’s ensemble expansion method outperforms both the full-schema baseline and other ablated variants, as shown in Table 4, with particularly strong gains on BIRD, where the schema becomes more challenging. We view this ensemble expansion as a strong contribution because it is fundamentally different from the schema selection strategies commonly used in state-of-the-art text-to-SQL systems. The challenge is greater in our data-synthesis setting, where, unlike standard text-to-SQL systems, we lack access to database contents and value-based retrieval mechanisms. Here we are trying to synthesize such data.
>
> Additionally, most cases where the gold query returns an empty result are not caused by low recall, but by schema ambiguities where multiple schema regions are semantically plausible. SynSQL selects one valid interpretation, while the gold query relies on another. These cases are still meaningful test databases because the data are aligned with the question and preserve schema integrity.
>
> ---
>
> ## Questions:
> > Could the authors elaborate on the qualitative difference between selection failures caused by the "w/o Expansion" variant and the full SynSQL variant?
>
> When the “w/o Expansion” variant fails, it’s often because there are several columns that mean almost the same thing. Since the model mainly relies on the natural language question, it tends to pick just one of those similar columns. Sometimes the ensemble or the “w/o Expansion” method selects only a single column even though more are needed. With the full ensemble + expansion approach, however, we can pull in those additional relevant columns, which leads to better overall performance.
>
> An illustrative example is the question: “What's the French name of the set of cards that ‘Tendo Ice Bridge’ is in?” In the ensemble phase, the system retrieves *foreign_data.name*, but the gold query actually uses *cards.name*. The expansion phase recovers this missing column because of its strong semantic similarity, ensuring that we do not overlook semantically similar columns that the LLM might not initially associate with the natural language question.
>
> > how can the framework be practically extended to ensure higher recall on complex, sparsely connected schemas without dramatically increasing the column count and sacrificing minimalism?
>
> This remains an active area of research in text-to-SQL. However, several practical extensions could improve robustness while maintaining the minimalist design principle. First, implementing multi-hop schema traversal guided by LLMs could recover essential join paths and connector tables in complex schemas, adding minimal columns while significantly boosting recall. Second, employing ensemble methods across multiple LLMs could reduce interpretation variance and yield more stable column predictions. These approaches offer promising directions for addressing the remaining failure cases while preserving SynSQL's core advantages. We have updated the paper in Sec. A.4 to include this path forward.
>
> ---
>
>
> We hope the detailed answers and updated sections in the paper address your concerns. We kindly ask you to consider the possibility of a score adjustment

---

### Author Response · Authors · 2025-11-28
**Overview of The Paper Updates**

Dear Meta-Reviewer,

We sincerely thank the reviewers for their valuable comments and feedback. In response, we have provided detailed clarifications and revised the paper accordingly. The updated version is available in the PDF above, with all modifications highlighted in blue for ease of reference.

---

The key updates are as follows:

- **New evaluation on newer datasets.** To address concerns regarding the reliance on older datasets (Spider and BIRD), we added experiments on the Spider 2.0-lite benchmark. The results show that SynSQL’s improvements persist on more challenging, newer datasets. See A.7.

- **Evaluation with stronger generator models.** In response to concerns about using weaker generator models, we included additional results using the stronger GPT-4.1 model. These results confirm that SynSQL consistently outperforms the vanilla baseline across all metrics. See A.6.

- **Experiments with state-of-the-art text-to-SQL systems.** To address concerns about using older text-to-SQL systems (DIN-SQL and DAIL-SQL), we incorporated evaluations with state-of-the-art systems from the BIRD leaderboard, Gemini-SQL, OmniSQL-32B, and CSC-SQL-32B, demonstrating that SynSQL’s benefits generalize across diverse model architectures. See A.8.

- **Clarification on extending the framework for higher recall.** To address questions about future extension of the framework to handle more complex schemas, we expanded the discussion of limitations and future work in A.4.

- **Analysis of diversification and the role of the Critic.** In response to concerns that LLMs may fail to diversify, we added a detailed analysis of the Critic component and clarified its importance. See A.5.

- **Discussion of related methods.** Following the suggestion to include comparisons with methods that generate synthetic natural language or SQL queries, we updated the related work section accordingly.

- **Additional clarifications.**
    - Schema reduction effects: Sec. 5.3
    - Details on the vanilla baseline: Sec. 5.1
    - Expanded description of auxiliary knowledge: Sec. 4.1

---

We believe these revisions and the clarifications below thoroughly address all questions and concerns raised by the reviewers. We appreciate your time and consideration in reviewing the updated manuscript and our responses.

---

### Meta-Review · Area_Chair_saLQ · 2026-01-01

**Summary:**

* Concern 1: potential issues with some components in the proposed test database generation system, such as omitting relevant tables/columns in the schema selector, or case-sensitivity (vdvq).
* Concern 2: The paper used relatively weak baselines (wjhd, wLfc), which could potentially make the evaluation results look more favorable as it's relatively easier to distinguish predicted queries from weak baselines against the ground-truth queries. wLfc also raised that the benchmarks (databases) used in the paper are not quite recent, and it's likely that the gaps between the proposed approach and existing baselines on synthesizing database contents could be narrower, given that it's more challenging to generate contents for databases with more complex schemas.
* Concern 3: Using LLMs as data generators might potentially limit the distribution of generated data as LLMs often lack enough output diversity (FQEo)

**Reviewer Concerns:**

Concern 1 and 3 are largely addressed. While the authors provided additional experiments on a more recent benchmark (Spider 2) and also using more recent text-to-SQL models on existing benchmarks used by the paper, Concern 2 still remains unsolved, since experiments with more recent text-to-SQL models show a drop on agreement rates (from 75%-80% reported in Table 1 to 65%~70%), and it is unclear whether this level of agreement rate is acceptable. More importantly, as wLfc commented on the new results, "The results on Spider 2.0 show that the magnitude of improvement decreases from double digits on Spider to about 4% which suggests on newer benchmarks, the relative advantage of the proposed method becomes substantially smaller, and it is unclear how robust the gains will be as benchmarks evolve." Therefore, I believe that the paper would benefit from another round of revision to address these oustanding concerns.

**Reviewer Scores:**

Reviewers FQEo and wjhd would have raised their scores as their concerns are largely addressed by the authors. However, as mentioned in reviser wLfc's follow-up reply to the authors' rebuttal, the additional experiment on Spider 2 remains concerning, so wLfc's score would have remained unchanged.

---

### Decision · Program_Chairs · 2026-01-26

Reject